# OpenReview forum: "Resource Consumption Red-Teaming for Large Vision-Language Models"
_ICLR.cc/2026/Conference — Submitted to ICLR 2026_

### Official Review · Reviewer_hxX8 · 2025-10-24

**Soundness:** 2
**Presentation:** 2
**Contribution:** 3
**Rating:** 6
**Confidence:** 4

**Summary:**

This paper introduces RECITE, a novel method for resource consumption attacks (RCA) on vision-language models (VLMs), leveraging Vision Guided Optimization. The study empirically demonstrates that RECITE can increase response latency by over 26 times and enhance GPU utilization by 20%. The experiments are extensive and well-conducted.

**Strengths:**

1. The paper proposed first method for exploiting visual input on RCAs.
2. The paper is well-structured and it is easy to follow.
3. The authors provide a comprehensive analysis, evaluating the attack's effectiveness across 3 VLMs.
4. The paper also include potential defense strategy, further enhancing their work.

**Weaknesses:**

1. The paper lacks experiments on transferability. The authors should evaluate the transferability of the optimized visual inputs, both across different white-box models and by using inputs optimized on a white-box model to attack a black-box model.
2. The paper has mentioned defense strategy. If diffusion purification or random smoothing were applied directly to the optimized inputs, would they have any defensive effect?

**Questions:**

See Weaknesses.

---

> ### Author Response · Authors · 2025-11-26
> **Rebuttal by Authors**
>
> We sincerely appreciate your valuable feedback and kind recognition of our discussion topic.
>
> ---
>
> ## **For Weakness 1**
>
> > **The paper lacks experiments on transferability. The authors should evaluate the transferability of the optimized visual inputs, both across different white-box models and by using inputs optimized on a white-box model to attack a black-box model.**
>
> We appreciate the reviewers' suggestions and have added transferability experiments, as shown in Table a and Table b. The increase in length after the attack is limited compared to the original input. We speculate that the visual encoders of different models disrupt the spatial structure of the perturbation. Therefore, improving transferability could be achieved through ensemble models. We plan to explore potential improvements in transferability in future work. The primary objective of this study is to elucidate how the activation of Output Recall can introduce substantial challenges in resource utilization.
>
> Table a:  The generation lengths in deployed LVLM services
>
> | Source\Target   | GPT-5.1 | GPT-4o-mini | Claude Sonnet 4.5 | gemini-2.5-flash |
> | --------------- | ------- | ----------- | ----------------- | ---------------- |
> | Original        | 13.4    | 74.7        | 121.4             | 126.7            |
> | Qwen2.5VL-7B    | 14.0    | 88.30       | 133.60            | 152.00           |
> | Llava1.5-7B     | 13.6    | 76.70       | 132.80            | 170.90           |
> | InstructBLIP-7B | 13.0    | 77.00       | 121.60            | 134.30           |
>
> Table b: Comparison of generation lengths in transferability experiments
>
> | Source\Target   | Qwen2.5VL-7B | Llava1.5-7B | InstructBLIP-7B |
> | --------------- | ------------ | ----------- | --------------- |
> | Original        | 91.2         | 40.0        | 39.34           |
> | Qwen2.5VL-7B    | 2046.9       | 40.9        | 80.5            |
> | Llava1.5-7B     | 97.8         | 2048.0      | 80.5            |
> | InstructBLIP-7B | 96.6         | 41.3        | 2030.7          |
>
> ## **For Weakness 2**
>
> > **The paper has mentioned defense strategy. If diffusion purification or random smoothing were applied directly to the optimized inputs, would they have any defensive effect?**
>
> RECITE adds perturbations to the input image, and thus image denoising methods are partially effective as a defense. We use random smoothing as a defense, optimize it with Gaussian filtering, set a 50% threshold for pixel changes, and test different smoothing radii.
>
> Test results for Qwen2.5 are shown in Table a. Compared to token consumption in RECITE attacks, the smoothing technique reduces total resource consumption, but the overall consumption remains above the average of the original samples. Compared to this input-side defense method, our proposed method provides defense at the generation stage. **The defense in our research is aimed at resource consumption targets, not limited to our attack method**. Although no text-based input or text-and-image hybrid attacks have yet been discovered, we hope our method can effectively mitigate this potential risk.
>
> Table c: Compare the effectiveness of different defense methods for RECITE attacks.
>
> | **Method**     | **Output Length** |
> | -------------- | ----------------- |
> | Original       | 39.38             |
> | RECITE-attack  | 2030.6            |
> | RECITE-defence | 11.7              |
> | Smoothing-1    | 81.1              |
> | Smoothing-2    | 100.3             |
> | Smoothing-3    | 105.3             |
>
> ---
>
> We sincerely look forward to further discussion to help clarify any remaining concerns. If you have additional comments or suggestions, we are committed to addressing them and continuously improving our work based on your feedback. **If you are satisfied with the revisions, we kindly hope you will consider increasing our score.**

---

> > ### Comment · Reviewer_hxX8 · 2025-11-26
> >
> > Thanks for your reply. The current rating is already positive, and I decide to keep it.

---

### Official Review · Reviewer_QQcr · 2025-10-29

**Soundness:** 3
**Presentation:** 3
**Contribution:** 3
**Rating:** 6
**Confidence:** 3

**Summary:**

This paper investigates visual-modality-driven resource consumption attacks (RCAs) on large vision-language models (LVLMs). The authors propose RECITE, a red-teaming framework that perturbs input images using an Output Recall Objective and Vision-Guided Optimization. The resulting perturbations are nearly imperceptible but cause the model to generate extremely long or looping outputs, significantly increasing GPU usage and latency. Experiments across multiple LVLMs demonstrate the feasibility of the attack and suggest preliminary defense strategies.

**Strengths:**

- The paper introduces a new attack surface—visual inputs causing resource consumption—which has not been systematically explored before. This problem is both novel and practically relevant.

- The proposed RECITE framework is simple yet effective, providing a structured way to red-team LVLMs for resource-related vulnerabilities.

- The experimental validation is extensive, involving multiple models and metrics (output length, GPU utilization, latency, memory). The results strongly support the main claim.

**Weaknesses:**

- The theoretical explanation of why visual perturbations cause looping behavior is insufficient. The paper would benefit from a formal analysis of the model’s stopping dynamics, such as EOS logit suppression or entropy evolution.

- The Output Recall Objective is largely heuristic. There is no ablation comparing it to simpler baselines such as minimizing the EOS token probability or tuning length penalties, which makes it unclear how necessary this specific objective is.

- The defense section is underdeveloped. The proposed sliding-window penalty lacks depth, and there is no quantitative analysis of how it affects model accuracy or normal captioning tasks.

- The evaluation scope is narrow, limited to open-source LVLMs. Testing on closed-source or API-based models (e.g., Gemini or GPT-4V) would make the results more compelling.

- The generality of the attack is uncertain. The paper does not explore transferability across models or tasks, nor does it test whether a single universal perturbation could generalize to multiple models.

**Questions:**

Have the authors considered testing on black-box, closed-source models to validate the practical impact in deployed systems?

---

> ### Author Response · Authors · 2025-11-26
> **Rebuttal by Authors (Part 1)**
>
> We sincerely appreciate your valuable and positive feedback to our paper.
>
> ---
>
> ## **For Weakness 1**
>
> > **The theoretical explanation of why visual perturbations cause looping behavior is insufficient. The paper would benefit from a formal analysis of the model’s stopping dynamics, such as** **EOS** **logit suppression or** **entropy** **evolution.**
>
> To clearly demonstrate RECITE's performance in attacks, we compare the output from eos logit and entropy evolution. Specifically, as shown in Figure a, the EOS logit of RECITE fluctuates less and remains consistently below 1e-03. This distribution ensures that **the EOS token is selected with a low probability under different adoption strategies**. Simultaneously, the confidence of the top-1 token in RECITE is higher than the normal output. By explicitly constraining the first n output tokens, we ensure that the output pattern remains stable throughout generation. From the entropy perspective, the RECITE pattern gradually stabilizes, and the entropy further decreases in the cyclic pattern (after an average of 10 tokens), indicating that the confidence of other tokens, including the eos token, is extremely low except for the cyclic target.
>
> We show a successful example of a single attack in Figure 2. It can be seen that although the EOS confidence is slightly higher than the normal output, its lower entropy value indicates that this stable pattern can be maintained in the long term.
>
> Table a: Model’s stopping dynamics in Qwen2.5-VL-7b.
>
> | Step | EOS Confidence ↑ | EOS Confidence ↑ | Top1 Confidence ↑ | Top1 Confidence ↑ | Entropy ↓ | Entropy ↓ |
> | ---- | ---------------- | ---------------- | ----------------- | ----------------- | --------- | --------- |
> |      | Original         | RECITE           | Original          | RECITE            | Original  | RECITE    |
> | 1    | 3.1e-06          | 1.1e-05          | 0.7               | 0.9               | 1.1       | 0.4       |
> | 3    | 4.5e-09          | 7.8e-10          | 0.7               | 0.9               | 0.6       | 0.4       |
> | 5    | 8.6e-09          | 6.4e-09          | 0.7               | 0.9               | 1.0       | 0.5       |
> | 7    | 0.1              | 3.6e-08          | 0.8               | 0.9               | 0.9       | 0.5       |
> | 9    | 2.7e-08          | 1.7e-06          | 0.4               | 0.9               | 2.2       | 0.6       |
> | 11   | 1.6e-07          | 1.6e-03          | 0.6               | 0.9               | 1.0       | 0.5       |
> | 13   | 4.3e-03          | 1.0e-03          | 0.5               | 1.0               | 1.9       | 0.2       |
> | 15   | 3.9e-03          | 9.3e-04          | 0.8               | 0.9               | 1.1       | 0.4       |
> | 17   | 5.9e-09          | 7.0e-04          | 0.7               | 0.9               | 1.1       | 0.3       |
> | 19   | 0.1              | 6.8e-04          | 0.7               | 1.0               | 0.9       | 0.2       |
> | 21   | 4.1e-09          | 5.9e-04          | 0.7               | 1.0               | 1.0       | 0.2       |
> | 23   | 7.8e-03          | 3.9e-04          | 0.7               | 1.0               | 1.3       | 0.2       |
> | 25   | 2.6e-10          | 7.6e-04          | 0.7               | 1.0               | 1.2       | 0.1       |
> | 27   | 2.4e-09          | 7.6e-04          | 0.7               | 1.0               | 0.8       | 0.1       |
> | 29   | 1.2e-09          | 3.7e-04          | 0.6               | 1.0               | 1.8       | 0.2       |

---

> > ### Author Response · Authors · 2025-11-26
> > **Rebuttal by Authors (Part 2)**
> >
> > Table b: A sample request for the model’s stopping dynamics in Qwen2.5-VL-7b.
> >
> > | Step | EOS Confidence ↑ | EOS Confidence ↑ | Top1 Confidence ↑ | Top1 Confidence ↑ | Entropy ↓ | Entropy ↓ |
> > | ---- | ---------------- | ---------------- | ----------------- | ----------------- | --------- | --------- |
> > |      | Original         | RECITE           | Original          | RECITE            | Original  | RECITE    |
> > | 1    | 2.9e-06          | 4.2e-05          | 0.9               | 0.9               | 0.6       | 0.4       |
> > | 3    | 3.8e-09          | 9.1e-10          | 0.6               | 0.7               | 0.7       | 0.6       |
> > | 5    | 3.0e-11          | 1.5e-08          | 1.0               | 1.0               | 0.0       | 0.1       |
> > | 7    | 5.6e-10          | 7.5e-10          | 1.0               | 1.0               | 0.0       | 0.1       |
> > | 9    | 1.8e-08          | 2.9e-08          | 0.6               | 0.8               | 1.2       | 1.0       |
> > | 11   | 1.3e-07          | 4.4e-07          | 0.5               | 1.0               | 1.1       | 0.2       |
> > | 13   | 3.8e-09          | 1.1e-06          | 0.7               | 1.0               | 1.0       | 0.1       |
> > | 15   | 9.0e-09          | 2.5e-06          | 0.8               | 1.0               | 0.8       | 0.1       |
> > | 17   | 2.6e-08          | 4.5e-06          | 0.7               | 1.0               | 1.7       | 0.1       |
> > | 19   | 0.6              | 1.7e-05          | 0.6               | 1.0               | 0.8       | 0.1       |
> > | 21   | -                | 5.4e-06          | -                 | 1.0               | -         | <0.05     |
> > | 23   | -                | 7.4e-06          | -                 | 1.0               | -         | <0.05     |
> > | 25   | -                | 1.9e-05          | -                 | 1.0               | -         | <0.05     |
> > | 27   | -                | 3.5e-06          | -                 | 1.0               | -         | <0.05     |
> > | 29   | -                | 4.8e-06          | -                 | 1.0               | -         | <0.05     |
> >
> > ## **For Weakness 2**
> >
> > > **The Output Recall Objective is largelyheuristic. There is no ablation comparing it to simpler baselines such as minimizing the EOS token probability or tuning length penalties, which makes it unclear how necessary this specific objective is.**
> >
> > As a key part of our contribution, we find it important to clarify the role of the Output Recall Objective. We compared RECITE with simpler baselines. Table c shows that traditional EOS suppression methods (Gao et al. ) [1] have some lengthening effect. Frequency penalties have a weaker impact on consumption attacks, with an effect not significantly different from that of benign requests. **Output Recall Objective effectively increases the average length by 2k, demonstrating its effectiveness as a target.**
> >
> > Table c: Comparison of output lengths from different methods. 'Gao et al.(original)' represents the experimental results in the paper [1].
> >
> > | Method                      | Benign | Attack |
> > | --------------------------- | ------ | ------ |
> > | Gao et al.(original)        | 54.4   | 131.8  |
> > | Gao et al. (500 iterations) | 39.4   | 42.7   |
> > | Frequency Penalty (0.8)     | 39.4   | 39.4   |
> > | Frequency Penalty (1.0)     | 39.4   | 35.6   |
> > | Frequency Penalty (1.2)     | 39.4   | 37.4   |
> > | RECITE                      | 39.4   | 2030.6 |
> >
> > [1] Gao, Kuofeng, et al. "Inducing High Energy-Latency of Large Vision-Language Models with Verbose Images." *The Twelfth* *International Conference on Learning Representations*.

---

> > > ### Author Response · Authors · 2025-11-26
> > > **Rebuttal by Authors (Part 3)**
> > >
> > > ## **For Weakness 3**
> > >
> > > > **The defense section is underdeveloped. The proposed sliding-window penalty lacks depth, and there is no** **quantitative analysis** **of how it affects model accuracy or normal captioning tasks.**
> > >
> > > We tested the performance impact of our defense using the general multimodal capability test dataset, MMMU [2]. The test results on Qwen are shown in Table d. **Our defense exhibits the same performance as the original model**. This indicates that our defense has a minimal impact on model usability. Since we dynamically suppress repetition semantics, this represents an improvement to the model's repetition penalty. It has little impact on normal text output and rarely triggers corrections due to the defense, so it does not alter the model's capabilities in most generation tasks.
> > >
> > > Table d: Impact of defense methods on RECITE’s performance. Comparison between the task completion success rate of the top five MMMU categories and the dataset-wide average.
> > >
> > > | **Task**                     | **Original** | **RECITE** |
> > > | ---------------------------- | ------------ | ---------- |
> > > | Accounting                   | 43.3%        | 43.3%      |
> > > | Agriculture                  | 53.3%        | 53.3%      |
> > > | Architecture_and_Engineering | 43.3%        | 43.3%      |
> > > | Art                          | 53.3%        | 53.3%      |
> > > | Art_Theory                   | 70.0%        | 70.0%      |
> > > | **Overall**                  | **45.9%**    | **45.9%**  |
> > >
> > > [2] Yue, Xiang, et al. "Mmmu: A massive multi-discipline multimodal understanding and reasoning benchmark for expert agi." *Proceedings of the IEEE/CVF Conference on* *Computer Vision and Pattern Recognition*. 2024.
> > >
> > > ## **For Weakness 4**
> > >
> > > > **The evaluation scope is narrow, limited to open-source LVLMs. Testing on closed-source or API-based models (e.g., Gemini or GPT-4V) would make the results more compelling.**
> > >
> > > We added experiments on closed-source based LVLMs, as shown in Table b. The horizontal axis represents the source model, and the vertical axis represents the target model.  The increased length of RECITE is limited. We speculate that the visual encoders of different models disrupt the spatial structure of the perturbation. Therefore, we plan to explore model ensembling in future work to further improve transferability.
> > >
> > > Table e:  The generation lengths in deployed LVLM services
> > >
> > > | Source\Target   | GPT-5.1 | GPT-4o-mini | Claude Sonnet 4.5 | gemini-2.5-flash |
> > > | --------------- | ------- | ----------- | ----------------- | ---------------- |
> > > | Original        | 13.4    | 74.7        | 121.4             | 126.7            |
> > > | Qwen2.5VL-7B    | 14.0    | 88.30       | 133.60            | 152.00           |
> > > | Llava1.5-7B     | 13.6    | 76.70       | 132.80            | 170.90           |
> > > | InstructBLIP-7B | 13.0    | 77.00       | 121.60            | 134.30           |

---

> > > > ### Author Response · Authors · 2025-11-26
> > > > **Rebuttal by Authors (Part 3)**
> > > >
> > > > ## **For Weakness 5**
> > > >
> > > > > **The generality of the attack is uncertain. The paper does not explore transferability across models or tasks, nor does it test whether a single universal perturbation could generalize to multiple models.**
> > > >
> > > > We recognize the importance of methodological universality in attack design. An effective approach should generalize across different model architectures and task categories. In our research, we discussed the **transferability of RECITE across different models** in Sec 4.2, showing that our method achieves an attack success rate of over 94% on three currently popular frameworks. Furthermore, **transferability across tasks** has been demonstrated in Appendix G. We attempted to build a general attack template in different tasks, exhibiting an attack success rate of over 50%.
> > > >
> > > > Regarding the transferability of cross-models, we have supplemented the experiments in Table f. The increase in length after the attack remains limited compared to the original input. As we clarified in Weakness 4, our primary goal is to demonstrate that the activation of Output Recall can cause substantial challenges in service resource utilization.v
> > > >
> > > > Table f: Comparison of generation lengths in transferability experiments
> > > >
> > > > | Source\Target   | Qwen2.5VL-7B | Llava1.5-7B | InstructBLIP-7B |
> > > > | --------------- | ------------ | ----------- | --------------- |
> > > > | Original        | 91.2         | 40.0        | 39.34           |
> > > > | Qwen2.5VL-7B    | 2046.9       | 40.9        | 80.5            |
> > > > | Llava1.5-7B     | 97.8         | 2048.0      | 80.5            |
> > > > | InstructBLIP-7B | 96.6         | 41.3        | 2030.7          |
> > > >
> > > > ## **For Questions 1**
> > > >
> > > > > **Have the authors considered testing on black-box, closed-source models to validate the practical impact in deployed systems?**
> > > >
> > > > For the answer to this question, please see Weakness 4 and Weakness 5.
> > > >
> > > > ---
> > > >
> > > > Some supplementary experiments have been completed on Qwen2.5-VL-7B, while those on other models are in progress. We will provide the results as soon as possible or include them in the final version if they are not ready in time.
> > > >
> > > > ---
> > > >
> > > > Thank you once again for your careful review. If our response has addressed your concerns, **we kindly request that you consider increasing our score.**

---

### Official Review · Reviewer_p9fs · 2025-11-01

**Soundness:** 2
**Presentation:** 2
**Contribution:** 2
**Rating:** 2
**Confidence:** 3

**Summary:**

This paper proposes RECITE (Resource Consumption Red-Teaming for LVLMs), a red-teaming framework that exploits visual inputs to trigger unbounded resource consumption attacks (RCAs) in large vision-language models (LVLMs). The core idea is to craft imperceptible adversarial perturbations that induce the model to enter a repetitive generation loop, e.g., outputting “cup cup cup...” indefinitely, thereby exhausting GPU memory and latency. Experimental results demonstrate that RECITE increases service response latency by over 26×.

**Strengths:**

1. The paper demonstrates that visual inputs alone can reliably trigger severe resource consumption attacks (RCAs) in large vision-language models (LVLMs).
2. The authors conduct extensive experiments across seven LVLMs from three major families (LLaVA, Qwen, BLIP), using diverse metrics (Output Time GPU Utilization Memory Usage) and multiple attack configurations.
3. The method section is technically thorough, with precise definitions of the Output Recall Objective and Vision Guided Optimization.

**Weaknesses:**

1. The claim that this is the “first” vision-based resource consumption red-teaming for LVLMs appears overstated. Prior work such as Gao et al. (ICLR 2024) [1] also leverages visual inputs to induce high latency/energy consumption in LVLMs. The paper should clarify how RE-CITE differs conceptually and technically from such approaches.

2. Figure 1, which depicts the RE-CITE pipeline, lacks sufficient clarity. Key components—such as visual encoding, embedding projection, and the iterative perturbation update process—are not well differentiated.

3. The core components demonstrates limited technical novelty. The Output Recall Objective seems to only define repetitive generation patterns and  the Vision Guided Optimization appear to be straightforward adaptations of the existing GCG method.

4. The evaluation only compares against GCG-RCAs. Given that [1] (Gao et al., ICLR 2024) also targets visual RCAs, it should be included as a baseline to better position RE-CITE’s relative effectiveness and novelty.

5. The paper does not assess whether perturbed images generated for one model transfer to others (e.g., perturbations optimized on LLaVA tested on Qwen). Such transferability is critical and should be evaluated.

6.  It remains unclear whether the generated perturbations are effective against deployed LVLM services (e.g., GPT4-V, Claude, or Qwen API)

[1] Gao K, Bai Y, Gu J, et al. Inducing High Energy-Latency of Large Vision-Language Models with Verbose Images[C]//The Twelfth International Conference on Learning Representations.

**Questions:**

Please address the weakness above.

**Details Of Ethics Concerns:**

.

---

> ### Author Response · Authors · 2025-11-26
> **Rebuttal by Authors (Part 1)**
>
> We sincerely thank you for your valuable comments. We will address the weakness in order to alleviate your concerns about our paper.
>
> ---
>
> ## **For Weakness 1**
>
> > **The claim that this is the “first” vision-based resource consumption red-teaming for LVLMs appears overstated. Prior work such as Gao et al. (ICLR 2024) [1] also leverages visual inputs to induce high latency/energy consumption in LVLMs. The paper should clarify how RE-CITE differs conceptually and technically from such approaches.**
>
> We appreciate the reviewer’s concern. In fact, **our research differs from previous work**, particularly from Gao et al. (ICLR 2024) [1], in the following aspects.
>
> | Perspective         | Gao et al. (ICLR 2024)                                       | RECITE                                                       |
> | ------------------- | ------------------------------------------------------------ | ------------------------------------------------------------ |
> | Target              | Extending the output length primarily to **increase resource consumption** | Achieving the maximum output window utilization to **maximize resource exhaustion** |
> | Perturbation Space  | Diverts the model’s attention to generate additional, irrelevant content | Disrupts semantic coherence, inducing infinite and self-repetitive outputs |
> | Method              | **Suppresses the end-of-sequence (EOS) token probability at each output step** | **Triggers the** ***Output*** ***Recall Objective*** **without depending on context** |
> | Evaluation Criteria | Total output length                                          | Number of outputs reaching the model’s maximum output window size |
> | Validity            | Produces extensions **under 1k tokens** (≈8× increase)       | 94% of outputs **reach the 2k token window limit** (≈26× increase) |
>
> Our method differs from previous studies in terms of objectives, techniques, and implementation results. Prior work [1, 2] primarily focused on manipulating the EOS token. While this approach offers targeted control, it inherently limits the scope of influence to content within the loss calculation's purview, making it challenging to consistently govern the broader generation pattern. In contrast, **RECITE operates directly by maliciously inducing repeated generation.** By satisfying the Output Recall Objective, our method ensures the model spontaneously produces extended, unbounded text.
>
> We hope this clarification helps convey RECITE's contributions more clearly. Should the reviewers find it helpful, we are willing to add this comparison to the appendix.
>
> [1] Gao, Kuofeng, et al. "Inducing High Energy-Latency of Large Vision-Language Models with Verbose Images." *The Twelfth* *International Conference on Learning Representations*.
>
> [2] Dong, Jianshuo, et al. "An Engorgio Prompt Makes Large Language Model Babble on." *The Thirteenth* *International Conference on Learning Representations*.
>
> ## **For Weakness 2**
>
> > **Figure 1, which depicts the RECITE pipeline, lacks sufficient clarity. Key components—such as visual encoding, embedding projection, and the iterative perturbation update process—are not well differentiated.**
>
> We will revise the manuscript to use a more precise workflow and to better highlight the key components, specifically emphasizing the following aspects:
>
> 1. **Visual encoding:** The blue and green arrows in the left of the Figure 1; we will replace them with clearer encoding modules.
> 2. **Embedding projection:** The embedding of the gray pixel matrix in the middle of the image; we will perform explicit corresponding embedding.
> 3. **Iterative update process:** The orange and black arrows at the top of the image; we will highlight the iterative process and the end of the discriminator.

---

> > ### Author Response · Authors · 2025-11-26
> > **Rebuttal by Authors (Part 2)**
> >
> > ## **For Weakness 3**
> >
> > > **The core components demonstrates limited technical novelty. The Output Recall Objective seems to only define repetitive generation patterns and the Vision Guided** **Optimization** **appear to be straightforward adaptations of the existing GCG method.**
> >
> > We adopted a similar perspective to GCG, yet the Vision-Guided Optimization in RECITE differs substantially in both its objective and mechanism. As the greedy algorithm is used by both GCG [4] and I-GCG [5], they employ parallel random token replacement and compare the similarity of the output content. The optimal value is then selected for the next round of iterative optimization. While effective in jailbreak attacks, **GCG remains difficult to implement using the** **Output** **Recall Objective**. The reasons are as follows:
> >
> > 1. GCG is mainly applied to jailbreak tasks, whose optimization process resembles that of classification problems and is therefore relatively easy. Our Output Recall Objective is a precise optimization goal, which disrupts the model's semantics and is inherently more difficult to optimize.
> > 2. GCG uses gradients for optimization rather than direct optimization, resulting in slower optimization.
> > 3. GCG uses a discrete convergence method, making it difficult to accurately find the convergence objective used in our research.
> >
> > In comparison, **RECITE directly backpropagates** **optimization** **gradients by leveraging image continuity, without requiring the parallel direction-selection procedure used in GCG**. As a result, RECITE is different from GCG and significantly more efficient. GCG consumes over 100× more computation time, as shown in Table 4.
> >
> > [4] Zou, Andy, et al. "Universal and transferable adversarial attacks on aligned language models." *arXiv* *preprint* *arXiv:2307.15043* (2023).
> >
> > [5] Jia, Xiaojun, et al. "Improved Techniques for Optimization-Based Jailbreaking on Large Language Models." *The Thirteenth* *International Conference on Learning Representations*.
> >
> > ## **For Weakness 4**
> >
> > > **The evaluation only compares against GCG-RCAs. Given that [1] (Gao et al., ICLR 2024) also targets visual RCAs, it should be included as a baseline to better position RE-CITE’s relative effectiveness and novelty.**
> >
> > Thanks to the reviewer for the supplement. We incorporated the method of Gao et al. (ICLR 2024) [1] into the baseline for comparison, and the results on InstructBLIP-7b are shown in Table a. The results show that all attack methods increase output length, and **our generation length is better than that of existing methods.**
> >
> > Because the demo by Gao et al. (ICLR 2024) [1] lacks an attack generation method in other models we use, we are migrating its code for testing. We will provide the results as soon as possible. Otherwise, we will add it to the final version.
> >
> > Table a： Comparison of output lengths from different methods. 'Gao et al.(original)' represents the experimental results in the paper [1].
> >
> > | Method                      | Benign | Attack | Extra consumption |
> > | --------------------------- | ------ | ------ | ----------------- |
> > | Gao et al.(original)        | 54.4   | 131.8  | 77.4              |
> > | Gao et al. (500 iterations) | 15.7   | 42.7   | 27.0              |
> > | RECITE                      | 15.7   | 2030.6 | 2014.9            |

---

> > > ### Author Response · Authors · 2025-11-26
> > > **Rebuttal by Authors (Part 3)**
> > >
> > > ## **For Weakness 5**
> > >
> > > > **The paper does not assess whether perturbed images generated for one model transfer to others (e.g., perturbations optimized on LLaVA tested on Qwen). Such transferability is critical and should be evaluated.**
> > >
> > > We tested the transferability of RECITE across three models, as shown in Table b. The horizontal axis represents the source model, and the vertical axis represents the target model. When the source and target models are the same, the RECITE method can effectively increase the generated length.
> > >
> > > Regarding transferability, the increase in length after the attack is limited compared to the original input. We speculate that the visual encoders of different models disrupt the spatial structure of the perturbation. Therefore, improving transferability could be achieved through ensemble models.
> > >
> > > In this paper, we reveal that the stable activation of Output Recall can lead to significant challenges in service resource consumption. In future work, we plan to improve the transferability of the proposed approach.
> > >
> > > Table b: Comparison of generation lengths in transferability experiments
> > >
> > > | Source\Target   | Qwen2.5VL-7B | Llava1.5-7B | InstructBLIP-7B |
> > > | --------------- | ------------ | ----------- | --------------- |
> > > | Original        | 91.2         | 40.0        | 39.34           |
> > > | Qwen2.5VL-7B    | 2046.9       | 40.9        | 80.5            |
> > > | Llava1.5-7B     | 97.8         | 2048.0      | 80.5            |
> > > | InstructBLIP-7B | 96.6         | 41.3        | 2030.7          |
> > >
> > > ## **For Weakness 6**
> > >
> > > > **It remains unclear whether the generated perturbations are effective against deployed LVLM services (e.g., GPT4-V, Claude, or Qwen API)**
> > >
> > > We added experiments for black-box deployed LVLM services. Table c shows that RECITE causes a slight increase in resource consumption in the black box. As clarified in Weakness 5, the current algorithm does not aggregate losses across multiple models. Our primary objective is to demonstrate that stable activation of Output Recall can pose substantial challenges for service resource utilization.
> > >
> > > Table c:  The generation lengths in deployed LVLM services
> > >
> > > | Source\Target   | GPT-5.1 | GPT-4o-mini | Claude Sonnet 4.5 | gemini-2.5-flash |
> > > | --------------- | ------- | ----------- | ----------------- | ---------------- |
> > > | Original        | 13.4    | 74.7        | 121.4             | 126.7            |
> > > | Qwen2.5VL-7B    | 14.0    | 88.30       | 133.60            | 152.00           |
> > > | Llava1.5-7B     | 13.6    | 76.70       | 132.80            | 170.90           |
> > > | InstructBLIP-7B | 13.0    | 77.00       | 121.60            | 134.30           |
> > >
> > > ---
> > >
> > > We have made our best effort to address identified issues. If there are remaining deficiencies, we would greatly appreciate any specific suggestions for improvement. **If you find that the revised manuscript meets the standards of a top-tier venue, we kindly ask you to consider increasing our score.**

---

### Official Review · Reviewer_tqRH · 2025-11-03

**Soundness:** 3
**Presentation:** 3
**Contribution:** 2
**Rating:** 4
**Confidence:** 3

**Summary:**

The paper proposes a denial-of-service (DoS) attack on vision-language models (VLMs). The attack identifies input samples that cause the model to consume an unusually large number of tokens, thereby degrading its efficiency. Such samples are discovered through a perturbation injection method guided by a newly proposed loss function.

**Strengths:**

- The problem studied is timely.

**Weaknesses:**

- The paper positions the work as a red-teaming effort, but the proposed method is more accurately described as a specific attack. In general, red-teaming involves systematically identifying a range of vulnerabilities, including those without concrete exploits, and typically provides comprehensive analysis and actionable recommendations. These broader aspects are missing from the current paper.

- Figure 3 measures semantic consistency, but its relevance to a denial-of-service and red-teaming  setting is not clearly justified.

- Although the attack is new in its application to VLMs, it largely follows existing adversarial attack paradigms. In my opinion,  the level of methodological novelty may not be sufficient for this conference..

**Questions:**

- Explain why the metric on "semantic consistency" and "feature similarity" are relevant.

- Including of actionable recommendations.

---

> ### Author Response · Authors · 2025-11-26
> **Rebuttal by Authors (Part 1)**
>
> We sincerely appreciate your timely response to our comments.
>
> ---
>
> ## **For Weakness 1**
>
> > **The paper positions the work as a red-teaming effort, but the proposed method is more accurately described as a specific attack. In general, red-teaming involves systematically identifying a range of vulnerabilities, including those without concrete exploits, and typically provides comprehensive analysis and actionable recommendations. These broader aspects are missing from the current paper.**
>
> We acknowledge the reviewer's conceptual clarification concerning red-teaming and attack. We fully concur with the insights provided on red teams, and we believe our research motivations align with the red-teaming paradigm.  **Our work extends beyond merely developing attack methodologies, further encompassing** **vulnerability** **disclosure, principled analysis, and mitigation strategies.**
>
> - In Section 5.1, we unveil a **novel** **vulnerability**: the model's generation logic, when broken by repetitive statements, can be driven into uncontrollable output. This type of proactive attack is distinct from model self-collapse and has not yet been systematically studied in existing literature. We conduct an attack based on this security vulnerability.
> - In Section 5.3, we analyze the **vulnerability****'s underlying principles**. By leveraging the output confidence distribution, we elucidate the causal mechanism of the infinite loop within RECITE.
> - In Section 5.4, we propose a targeted **dynamic mitigation strategy** that reduces attack resource consumption by over 50%.
>
> Consequently, our research is positioned not merely as an attack, but as a comprehensive red-teaming endeavor. **If you consider it appropriate, we would be glad to adjust the paper’s structure** to better emphasize our contributions to vulnerability disclosure and principled analysis, aligning it more closely with the conventions of red-team research.
>
> ## **For Weakness 2**
>
> > **Figure 3 measures semantic** **consistency****, but its relevance to a denial-of-service and red-teaming setting is not clearly justified.**
>
> Figure 3 employs semantic consistency to quantify attack stealth, which we assert is a critical and widely recognized criterion in safety research. **The pervasive use of stealth metrics across diverse domains,** such as adversarial [1,2], jailbreak [3,4], and resource-exhaustion attacks [5], underscores their importance for evaluating attack effectiveness.
>
> In our work, high semantic consistency indicates that the attack does not visibly alter the image’s meaning from a human perspective, thereby bypassing detection by maintenance personnel. This inherent stealth is precisely what **makes malicious attacks significantly more challenging to identify and mitigate in practical, real-world scenarios**.
>
> [1] Zhang, Chaoning, et al. "Cd-uap: Class discriminative universal adversarial perturbation." *Proceedings of the AAAI conference on artificial intelligence*. Vol. 34. No. 04. 2020.
>
> [2] Benz, Philipp, et al. "Double targeted universal adversarial perturbations." *Proceedings of the Asian Conference on* *Computer Vision*. 2020.
>
> [3] Mu, Honglin, et al. "Stealthy jailbreak attacks on large language models via benign data mirroring." *Proceedings of the 2025 Conference of the Nations of the Americas Chapter of* *the Association for Computational Linguistics**: Human Language Technologies (Volume 1: Long Papers)*. 2025.
>
> [4] Geng, Jianing, et al. "When Safety Detectors Aren't Enough: A Stealthy and Effective Jailbreak Attack on LLMs via Steganographic Techniques." *arXiv* *preprint* *arXiv:2505.16765* (2025).
>
> [5] Gao, Kuofeng, et al. "Inducing High Energy-Latency of Large Vision-Language Models with Verbose Images." *The Twelfth* *International Conference on Learning Representations*.
>
> ## **For Weakness 3**
>
> > **Although the attack is new in its application to VLMs, it largely follows existing adversarial attack paradigms. In my opinion, the level of methodological novelty may not be sufficient for this conference.**
>
> We would like to clarify our main contributions in this paper. First, it is important to emphasize that **our work extends beyond merely constructing an attack method**. This study formally characterizes a novel class of resource consumption vulnerabilities and establishes a robust, stable triggering mechanism. Distinct from incidental model crashes induced by benign inputs or existing End-of-Sequence (EOS) suppression techniques, the vulnerability detailed in Section 3.1 provides a formal, generalized attack structure that demonstrably induces severe resource consumption across any input request.
>
> Second, RECITE diverges from existing adversarial attacks, as **its target shifts from typical misclassification to inducing unbounded generation**. To effectively address this novel attack paradigm, we introduce the Output Recall objective, specifically formulated to induce unbounded outputs.

---

> > ### Author Response · Authors · 2025-11-26
> > **Rebuttal by Authors (Part 2)**
> >
> > ## **For Questions 1**
> >
> > > **Explain why the metric on "semantic** **consistency****" and "feature similarity" are relevant.**
> >
> > As explained **in our response to Weakness 2**, we clarify the necessity of considering “semantic consistency” in our research.
> >
> > The original image and the RECITE sample maintain a high degree of consistency in the semantic space, with no shift in core semantic labels, making them difficult for humans to detect. In terms of visual features (such as color, texture, and structure), their features overlap, and perturbation does not disrupt feature expression. They are also difficult for human observers to detect. Thus, "semantic consistency" and "feature similarity" are correlated.
> >
> > ## **For Questions 2**
> >
> > > **Including of actionable recommendations.**
> >
> > As clarified **in our response to Weakness 1**, we propose an actionable recommendation in Section 5.4 that can reduce attack resource consumption by more than 50%.
> >
> > ---
> >
> > If our response has addressed your concerns, **we kindly request that you consider increasing our score.** Thank you once again for your careful review, and we look forward to further discussions with you.

---

> ### Comment · Reviewer_tqRH · 2025-11-26
>
> Thanks for the response.
>
> (1) Weakness 1.
>
> I feel that the explanation still revolves around a specific attack instead on red-teaming methodology.  There is a fundamental difference between red teaming and specific attack.   Red teaming's goal is to help in the defense.
>
> In the context of cybersecurity, a work that proposes a new attack (e.g. a new way of buffer-overflow, or flaw in TLS re-negotiation protocol) is  generally credited as a novel attack or new vulnerability, and not considered as a contribution to red-teaming methodology.      On the other hand, works that enhance systematic exploration could be considered closer to red-teaming methodology.   Although the paper highlighted that its contribution is on red-teaming methodology, IMHO, this paper should be classified as contribution of a more effective attack.
>
> I believe it is very important to align the terminologies to current practice, so as avoid misunderstanding among practitioners and researchers in the community.

---

> > ### Author Response · Authors · 2025-11-27
> >
> > ## For Reply
> >
> > We sincerely thank you for the detailed feedback and constructive suggestions. Your comments have helped us better clarify the focus of our work. **We will revise the manuscript accordingly, shifting the emphasis from red-team methodologies to vulnerability analysis and more effective attack strategies.**
> >
> >  We would greatly appreciate any further specific suggestions for improvement. If the revised version meets the standards expected of a top-tier venue, we kindly ask you to consider an appropriate score adjustment.

---

### Meta-Review · Area_Chair_krqo · 2026-01-11

**Summary:**

This paper introduces RECITE, a red-teaming framework that exposes a new class of resource consumption attacks in large vision-language models by exploiting visual inputs. The method uses fine-grained, pixel-level adversarial perturbations to induce repetitive, unbounded generation, leading to significant increases in response latency, GPU utilization, and memory consumption.

During the rebuttal, the authors partially addressed the reviewers’ concerns, but several major issues remain. First, as pointed out by Reviewer tqRH and acknowledged by the authors, the focus of this work should not be on red-teaming methodology but rather on vulnerability analysis and more effective attack strategies, which would require a substantial restructuring of the paper. Second, as a single attack method, the approach lacks strong transferability and cannot be used as a black-box attack on commercially deployed models, as raised by multiple reviewers. This further weakens the contribution of the work to the community. Third, the proposed defense is technically simple, as noted by Reviewers QQcr and hxX8.

Taken together, we recommend rejecting this work.

**Reviewer Concerns:**

Reviewers QQcr (score: 6) and hxX8 (score: 6) have most of their main concerns addressed.

Reviewer tqRH’s (score: 2) concerns regarding the overemphasis on red-teaming and the limited contribution as a single attack method remain.

Reviewer p9fs’s (score: 4) concerns regarding transferability, the black-box setting, and the defense mechanism also remain unaddressed.

**Reviewer Scores:**

Reviewers QQcr (score: 6) and hxX8 (score: 6)  would maintain their score.

Reviewer tqRH’s (score: 2) could potentially increase the score as the authors have addressed some of the concerns but remain negative.

Reviewer p9fs’s (score: 4) would not increase the score as most of the concerns are not addressed.

---

### Decision · Program_Chairs · 2026-01-26

Reject